# Complex multidisciplinary intervention to improve Initial Medication Adherence to cardiovascular disease and diabetes treatments in primary care (the IMA-cRCT study): mixed-methods process evaluation protocol

Carmen Corral-Partearroyo,[1,2] Alba Sánchez-Viñas,[1,3] Montserrat Gil-Girbau [iD],[1,4] Maria Teresa Peñarrubia-María,[1,5] Ignacio Aznar-Lou,[1,6] Carmen Gallardo-González,[1,5] María del Carmen Olmos-Palenzuela,[7] Maria Rubio-Valera[1,6]

For numbered affiliations see end of article.

**Correspondence to**
Dr Montserrat Gil-Girbau;
mariamontserrat.gil@sjd.es

## ABSTRACT

**Introduction** Medication non-initiation, or primary non-adherence, is a persistent public health problem that increases the risk of adverse clinical outcomes. The initial medication adherence (IMA) intervention is a complex multidisciplinary intervention to improve adherence to cardiovascular and diabetes treatments in primary care by empowering the patient and promoting informed prescriptions based on shared decision-making. This paper presents the development and implementation strategy of the IMA intervention and the process evaluation protocol embedded in a cluster randomised controlled trial (the IMA-cRCT) to understand and interpret the outcomes of the trial and comprehend the extent of implementation and fidelity, the active mechanisms of the IMA intervention and in what context the intervention is implemented and works.

**Methods and analysis** We present the protocol for a mixed-methods process evaluation including quantitative and qualitative methods to measure implementation and fidelity and to explore the active mechanisms and the interactions between the intervention, participants and its context. The process evaluation will be conducted in primary care centres and community pharmacies from the IMA-cRCT, and participants include healthcare professionals (general practitioners, nurses and community pharmacists) as well as patients. Quantitative data collection methods include data extraction from the intervention operative records, patient clinical records and participant feedback questionnaires, whereas qualitative data collection involves semistructured interviews, focus groups and field diaries. Quantitative and qualitative data will be analysed separately and triangulated to produce deeper insights and robust results.

**Ethics and dissemination** Ethical approval has been obtained from the Research Ethics Comittee (CEIm) at IDIAP Jordi Gol (codeCEIm 21/051 P). Findings will be disseminated through publications and conferences, as well as presentations to healthcare professionals and stakeholders from healthcare organisations.

## STRENGTHS AND LIMITATIONS OF THIS STUDY

⇒ This process evaluation will explain how the intervention was implemented, how different components interact and work and how they influence outcomes.

⇒ This study includes a wide range of quantitative and qualitative research methods; it is logistically challenging and time consuming. A multidisciplinary research team has been involved.

⇒ The flexible and pragmatic design will be crucial to react to changes and adapt the intervention to emerging contextual factors.

⇒ Data collection methods have been designed to adapt to the participants in what we anticipate might be an overloaded and difficult time due to the persisting COVID-19 pandemic.

⇒ There is a risk of response bias among professionals that answer questionnaires and agree to participate in the qualitative evaluation as they may have engaged more with the intervention. Additionally, patients will be recruited by professionals and this might bias their responses and the decision of the patient towards filling the prescription.

**Trial registration number** NCT05026775.

## INTRODUCTION

Medication non-initiation, or primary non-adherence, is defined as not initiating the prescribed pharmacological treatment.[1] In recent years, there has been an increase in evidence regarding non-initiation.[2–5] It is subject to patients' characteristics and motivations, the pharmacological treatment prescribed and the context,[4 6 7] and for some treatments, it reaches a prevalence of 40%.[3]

Adherence to long-term medications has been shown to be crucial to the prevention of further complications.[8] Low adherence to cardiovascular disease (CVD) and diabetes treatments worsens patients' clinical outcomes[9–12] and increases direct and indirect costs to healthcare systems,[10 13 14] highlighting the need for interventions to prevent it.

In the past, some studies evaluated the effectiveness of interventions to improve non-initiation, focused mainly on CVD medications.[15–20] The majority were based on patients' reminders: two on automated messages,[15 19] two on phone calls performed by professionals[17 18] and one on both automated and professional's phone calls.[16] Only two of these studies reported a significant decrease in non-initiation,[15 19] and most showed a high overall risk of bias. Hawthorne effect and desirability bias was high overall due to lack of blinding of participants and the characteristics of the outcome under study[15–18 20]; most studies used medicine acquisition as a proxy for initiation with no further follow-up, and false-positive initiation could occur when patients know they are being observed.[21 22] None of the interventions tested was described as being founded on a health behaviour change theory.

In the last decade, there has been growing interest in behavioural interventions based on shared decision-making (SDM) to improve adherence.[23–26] SDM is a process whereby the professional and the patient jointly decide on a treatment or healthcare choice.[27] Both share their knowledge, and the patient is invited to express their preferences and consider all options to achieve a mutual agreement.[27 28] This respects patient autonomy yet offers guidance to the patient by involving them in the decision at their preferred level.[27] By involving the patient in the decision process, SDM increases patients' health literacy and satisfaction.[23–26 29] However, there is not sufficient evidence for an effect of SDM-based interventions on medication adherence, and there is a lack of standardised outcomes in studies evaluating the impact of SDM interventions on adherence to pharmacological treatments.[23–26 29]

## The non-initiation project

The non-initiation project is based on the framework for developing and evaluating complex interventions proposed by the Medical Research Council (MRC)[30 31] and aims to develop and evaluate an intervention to decrease non-initiation. Figure 1 summarises the project phases.

In phase I, or the development phase, the prevalence and explanatory factors of non-initiation were explored. Overall prevalence of non-initiation in primary care (PC) in Catalonia (Spain) was found to be 18% and between 6% and 9% for treatments for CVD and diabetes.[5 32] Predictors of non-initiation included patient characteristics (such as being younger), the treatment (such as cost) and the system (such as receiving the prescription from a substitute or resident general practitioner (GP)).[5 32] The patients' reasons for non-initiation were explored by carrying out qualitative research with patients and professionals.[7 33] Based on the results of these studies, the Initial Medication Adherence (IMA) intervention, a complex, multidisciplinary intervention to improve initiation and adherence to CVD and diabetes treatments, was modelled. To increase the acceptability of the intervention, discussion groups were conducted with GPs, nurses, pharmacists, social workers, cardiologists, endocrinologists and internists, who made suggestions for refinement, described its limitations and anticipated barriers to its implementation.

To assess the feasibility of the IMA intervention and the evaluation design, a pilot trial with an embedded process evaluation was conducted as part of phase II, or feasibility phase (ClinicalTrials.gov, NCT05094986). Detailed methods and results of the pilot study are presented elsewhere.[34] The intervention components and implementation strategies were considered feasible and acceptable. However, barriers to the engagement of professionals, training for professionals and intervention decision aids

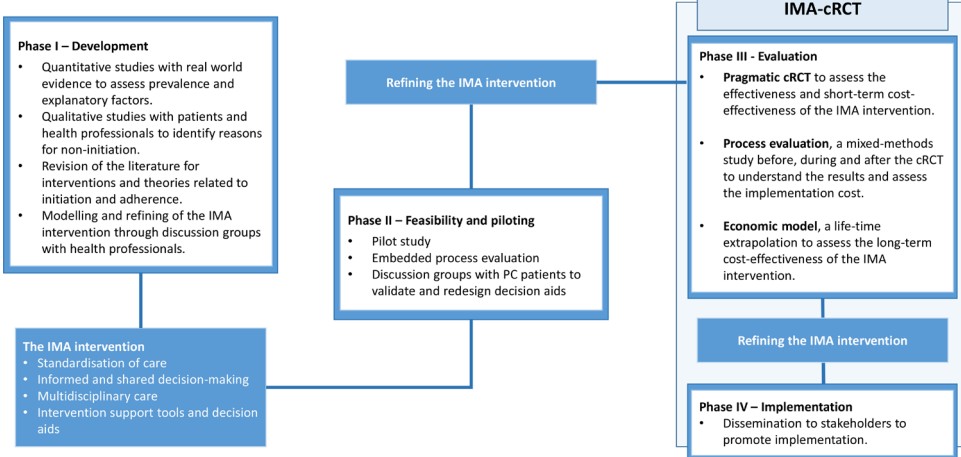

**Figure 1** IMA intervention phases: development, feasibility, evaluation and implementation. cRCT, cluster randomised controlled trial; IMA, initial medication adherence; PC, primary care.

were identified. These results were used to refine the IMA intervention prior to the definitive cluster-randomised controlled trial (cRCT).

The process evaluation outlined in this paper is integrated into the IMA-cRCT, phase III or evaluation phase: a pragmatic cRCT with two parallel groups that aims to evaluate the effectiveness, cost-effectiveness and understand the impact of the IMA intervention. Detailed cRCT methods (ClinicalTrials.gov, NCT05026775) are described elsewhere.[35] The trial is being conducted in 24 PC centres in Catalonia (May 2022–September 2023), randomised to the control (usual care) or the intervention group (the IMA intervention), as well as community pharmacies in the area covered by PC centres of the intervention group. Professionals in the intervention group were trained on the IMA intervention and will apply it to all patients receiving a new prescription for lipid-lowering medication, antihypertensive medication, antiplatelet medication and/or antidiabetic medication during the study period (7 months).[35] The primary outcome of the trial is the rate of initiation. Secondary outcomes include other measures of adherence (implementation and persistence), clinical outcomes and cost-effectiveness.

## The IMA intervention

The IMA intervention is founded on the theoretical model for non-initiation.[7 33] According to this model, the decision to initiate pharmacological treatments is multifactorial, and it is influenced by the patients' beliefs about the disease and treatment options, the existence of non-pharmacological measures, the interaction with healthcare professionals (GPs, nurses and pharmacists) and the context, cultural factors and health literacy of the patient.[7 33] The model suggests that an intervention that improves health literacy, helping the patient to understand the risks of the disease and the benefits and risks of treatment options and involves the patient in the decision-making process could improve initiation and long-term adherence.[7 33] The model also highlights the influence of healthcare professionals and the importance of multidisciplinary recommendations when a new pharmacological treatment is prescribed.

As illustrated by the non-initiation model, the IMA intervention is expected to work at the intrapersonal level by increasing patients' health literacy and empowerment and the interpersonal level by promoting SDM through the interaction between the patient and healthcare professionals and supporting the standardisation of clinical practice among all the PC professionals that interact with the patient (GPs, nurses and pharmacists).

During a consultation, the GP applies the principles of SDM.[27 28] They define the problem and decision at hand by providing information about the disease and treatment options and exploring the patient's perspectives, concerns and expectations supported by decision aids. Both the GP and patient have coresponsibility to negotiate a decision before the prescription of a new CVD or diabetes pharmacological treatment is issued. When

necessary, the decision is delayed to offer the patient the opportunity to reflect on it, obtain complementary information (reliable decision aids are recommended as sources of information) and/or discuss the decision with others (including nurses and pharmacists). When consulted by patients, nurses and pharmacists explore patients' queries regarding new CVD or diabetes medication prescriptions, or those of patients considering the use of medication, and use decision aids to provide information support, standardising the message from all healthcare professionals and improving interdisciplinary collaboration. In the case that the patient changes their mind about the use of medication, nurses and pharmacists refer them back to the GP. The IMA intervention is a one-shot intervention at the time of a new prescription. The dosage, or times the intervention has been applied to the same patient, varies on the healthcare professionals (GPs, nurses and pharmacists) consulted during and after a new prescription and whether they are participating in the trial, with the minimum dose being one time (when the prescription is issued).

The logic model illustrated in figure 2 shows how the intervention would primarily influence the adequate use of treatment (primary and secondary adherence) and ultimately impact the health outcomes of the population under study, as well as influence the interdisciplinary collaboration between professionals and patient–healthcare professional interaction. The IMA intervention has three main inputs as part of the implementation strategy (figure 2). First, *professional engagement* increases professionals' interest and promotes participation. PC and pharmacy stakeholders, including scientific organisations, healthcare quality agencies, official colleges, and managers and directors of PC centres were first contacted and informed. Thereafter, professionals were informed at PC centres, community pharmacies and official colleges. Second, the IMA intervention *training* was provided to professionals (GPs, nurses and pharmacists) in two sessions of 3 hours each. Professionals were trained together to promote standardisation and mutual understanding of each other's role and to generate bonds. The first session covers the basics of the intervention: the evidence on non-initiation, the practical aspects of the intervention, the role of each professional and the intervention decision aids. The second session was designed by an SDM expert and focuses on SDM and communication skills. Third, the IMA intervention *decision aids* promote discussion of all relevant topics with the patient and SDM (increasing adherence to the intervention and standardisation of practice). *Leaflets* (one for each pharmacotherapeutic group) contain information on the risks of the disease, the risks and benefits of pharmacological and non-pharmacological treatments and key messages to encourage the patients to express their opinions and share their uncertainties with the professional, as well as other reliable sources of information (including other healthcare professionals and a website). The *website* www.iniciadores.es is divided into pathologies and

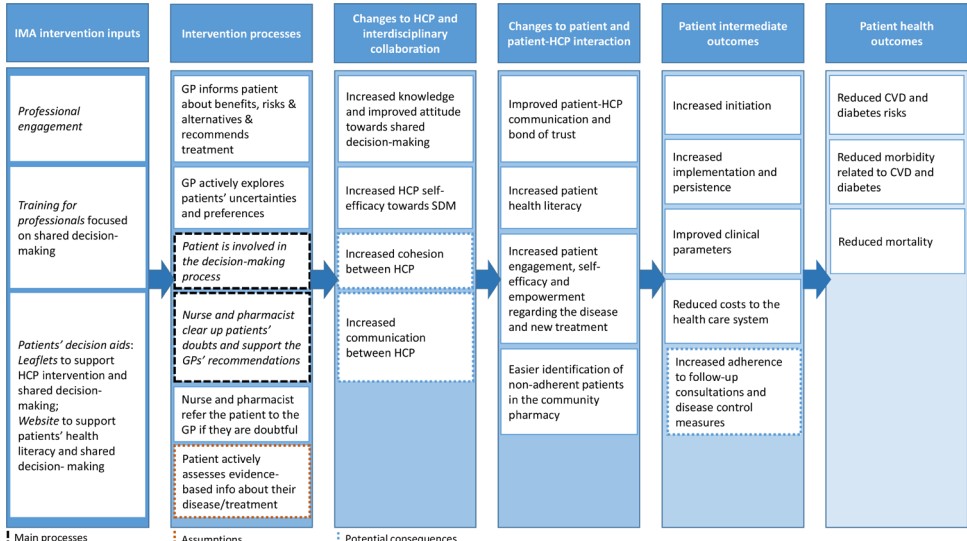

**Figure 2** IMA intervention logic model. CVD, cardiovascular disease; GP, general practitioner; HCP, healthcare professional; IMA, Initial Medication Adherence.

pharmacological treatments, with extended information on the disease, treatments and additional links to other reliable websites (such as those run by the national health system). The content of the leaflets and website are reliable and are endorsed by public healthcare organisations.

### The IMA intervention in the context of the COVID-19

In the case of a COVID-19 outbreak, when a new treatment is recommended during a telephone consultation, the doctor sends the leaflet through email and refers the patient to the website (where leaflets are also available). Additionally, the patient can collect the leaflet from participating pharmacies when collecting the medication. In this case, a follow-up telephone consultation (GP or nurse) is recommended a week after the prescription is issued to explore patients' queries and concerns.

### Process evaluation

Randomised controlled trials (RCTs) have been presented as the gold standard for evaluating effectiveness and efficiency of complex interventions.[36] Complex interventions combine multiple components that interact with each other, involve several stakeholders and generally require a behavioural change by those that implement and receive the intervention.[30 31 36] However, RCTs typically have rigid designs, tend to focus mainly on outcome effect and fail to explain how the intervention was implemented and in what context, what the active components were and for whom it worked.[30 31] Process evaluations embedded in pragmatic RCTs are needed to understand how the intervention was delivered, how different components interact and work, how they influence the intervention's primary and secondary outcomes and its effectiveness.[37]

Ultimately, some very efficient interventions can be difficult to translate into routine practice, especially when the intervention cost is high because it requires organisational and behavioural changes. Assessing the cost of implementation, costs of the strategies to put in practice

and sustain an intervention, provides decision makers with relevant information when evaluating the translation of the intervention into routine clinical practice.[38]

### Aims and objectives

This process evaluation aims to understand the implementation and mechanism of action of the IMA intervention and how the context affects them and therefore understand and explain the results of the cRCT in terms of effectiveness and cost-effectiveness, refine the IMA intervention and provide information on replicability and generalisability to other contexts.

The objectives of the study are to:
1. Assess the extent to which the IMA intervention was implemented as intended (fidelity) and understand how the IMA intervention becomes integrated into routine healthcare practice (implementation).
2. Identify and understand the active mechanisms of the IMA intervention (mechanisms of impact).
3. Understand the context where the IMA intervention is implemented and identify factors that can influence the IMA intervention's active mechanisms (context).
4. Assess the cost of implementing the IMA intervention.

### METHODS AND ANALYSIS
### Study design and framework of the process evaluation

A mixed-methods process evaluation study will be undertaken, involving analysis of real-world practice evidence, data collection forms, field diaries and interviews with professionals and patients. The MRC guideline for process evaluations of complex interventions was used to guide the design of this evaluation.[37] It focuses on three domains that interact with each other: (1) the implementation of the intervention; (2) the mechanisms of the intervention that affect the outcomes; and (3) the characteristics of the contexts that can influence the previous domains in the intervention group and control group

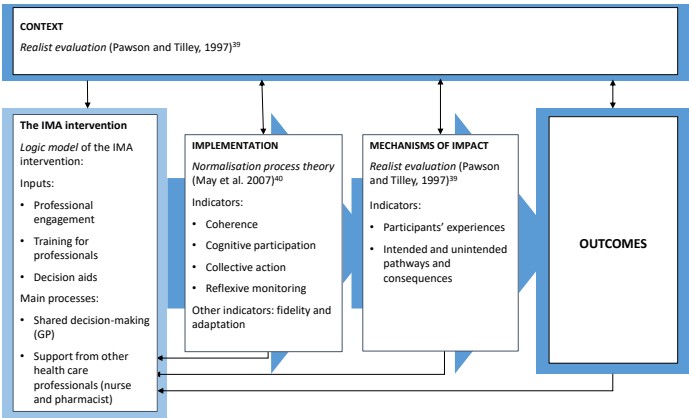

**Figure 3** Domains and theoretical framework of the IMA-cRCT process evaluation (adapted from Moore *et al*[37]). GP, generalpractitioner; IMA, Initial Medication Adherence.

contextual factors that could influence the outcomes of the cRCT.

Figure 3 shows the interaction between the three domains and the theoretical frameworks used to evaluate them.[39 40] Implementation will be assessed through the normalisation process theory,[41 42] which explains how an intervention becomes routinely integrated into everyday healthcare practice by assessing four indicators: coherence, cognitive participation, collective action and reflexive monitoring.[42] The evaluation of the mechanisms of impact and the context of the intervention will be assessed following the recommendations of realist evaluation[39] to explain how active mechanisms of the intervention generate an effect, for whom and under what circumstances.[37 39] It takes into account the expected and unexpected consequences that may result from the activation of different hypothesised mechanisms in different contexts and end up generating different effects.[37 39]

Additionally, the Stages of Implementation Completion framework will be used, and the Cost of Implementing New Strategies tool will be adapted to the IMA intervention[43] to assess the costs of implementation when translating the IMA intervention into other settings.

The process evaluation will be conducted by the IMA-cRCT research team. This is a multidisciplinary team formed by researchers with expertise in quantitative and qualitative research and PC professionals. The main components of the intervention are not expected to be adapted during the trial, although the dynamic approach of this pragmatic evaluation gives the design the flexibility to react to changes if needed.

### Setting and participants
The IMA intervention will be implemented in PC centres and community pharmacies in Catalonia (Spain). PC is the gateway to the healthcare system and where the vast majority of the prescriptions for long-term treatments are issued.[44] Patients have an assigned GP and nurse. GPs monitor and prescribe treatments and nurses follow-up the patient. Medications are electronically prescribed, and patients can fill them only at community pharmacies where the pharmacist can check the prescription directly through the electronic prescription system.[44] A warning appears when a new platelet aggregation inhibitors excluding heparin and insulin is going to be dispensed. Full details on the cRCT setting are provided elsewhere.[35]

The IMA-cRCT will recruit 24 PC centres, around 300 professionals and 4000 patients.[35] The sampling strategy of the process evaluation is conditioned by the cluster sampling design of the cRCT and is detailed below.

### Intervention group
#### Professionals
GPs, nurses and pharmacists from all PC centres will have feedback questionnaires sent to them, but only professionals from six to eight PC centres will be recruited to participate in the qualitative research due to time restrictions. The recruitment will be based on a theoretical sampling strategy according to the type of PC centre (size and location) and taking into account predictors of non-initiation (socioeconomic level of the area, rurality and the proportion of immigrant population). Professionals from the recruited PC centres will be invited to participate in the interviews by phone calls following maximum variation sampling according to role and type of contract, years of experience in PC, sex and age, nationality and owner or not of the pharmacy.

Each PC centre will have a study coordinator, a GP or nurse. The coordinator will be the link with the research team, promoting the implementation of the intervention, coordinating the distribution of intervention materials and informing the research team of external events that may influence the correct implementation of the intervention and development of the trial.

*Patients*

The coordinator of the PC centres will identify patients that were prescribed a new treatment for CVD or diabetes during the study period and invite them to participate in an interview. They will be asked to follow a variation sampling strategy based on medication prescribed and treatment being initiated or not. In addition, we will follow a maximum variation criterion based on the previous aspects, educational and socioeconomic level, sex and age. Once patients agree to participate, the research team will contact them by phone calls to provide further information about the study.

### Control group

The coordinator from the control group PC centres will be contacted and interviewed by phone calls to explore any external events and contextual factors that could have influenced outcomes on the usual care groups and, therefore, the results of the cRCT.

### Data collection

The specific objectives and research questions, as well as the data collection method used to assess the first three domains are presented in table 1 (implementation), table 2 (mechanisms of impact) and table 3 (context). Specific quantitative and qualitative methods used to meet the process evaluation aims and objectives are described for each domain.

### Implementation

Data will be collected on intervention fidelity to identify how consistent the implementation of the intervention was with the initial plan and if it required any adaptations during the trial, as well as assess the implementation into routine practice (table 1). Fidelity will be assessed through quantitative data on professional interaction and the intervention implementation plan (PC centres and professional engagement, training attendance, use of intervention tools and follow-up consultations). Adaptations will be assessed using quantitative data from professionals' feedback questionnaires and qualitative data from the coordinator's field diary.

Additionally, qualitative methods will be used to evaluate the implementation of the IMA intervention into routine PC centre practice. Interviews with professionals will assess the perceived need and adequacy of the IMA intervention as well as measures used to appraise it. Professionals' feedback questionnaires will collect data on professionals' attitudes towards the IMA intervention before and after the trial and how it is operationalised and integrated into routine practice.

### Mechanisms of impact

Qualitative methods will be used to identify and understand the active mechanisms of the IMA intervention that bring about any effects and explains the intervention's logic (table 2). Interviews with professionals and patients will explore their perspectives and experiences with the intervention, potential changes to professionals' attitudes and interdisciplinary collaboration, changes to patients' knowledge, behaviour and interaction with professionals and any expected or unexpected consequences.

### Context

Data on the context of both the intervention and control groups will be collected (table 3). Demographic data from the PC centres will be extracted from Catalan health system records. Interviews will be carry out with professionals and patients to explore the context of the PC centre and examine in which circumstances mechanisms of impact work and therefore influence the study outcomes.

### Cost of implementation

To assess the cost of implementing the intervention, all human and material resources used in each stage of the implementation process to put the IMA intervention into practice will be collected and taken into account.

### Data collection methods

#### Quantitative methods

*Monitoring data*

Data will be collected from the operative records, website records and clinical records from real-world databases in the public PC system in Catalonia (System for the Development of Research in Primary Care).[45] Data will be structured and descriptively summarised to assess fidelity, context and cost of implementation through:

► Professional engagement: number of PC centres and professionals that decline to participate after the information session.
► Training attendance rate.
► Intervention tools usage rate: website indicators (number of views, percentage of rebound, mean view time and depth) and number of times the leaflet was downloaded.
► Follow-up consultation rate at the PC centre after a new prescription.
► Demographic records: PC centre size and location, number of professionals, socioeconomic level of the area, rurality, average age of the population and the proportion of the immigrant population.
► Implementation costs: human resources based on time invested and professional category, and consumable materials based on units used.

*Professionals' questionnaires*

Professionals will be asked to complete post-training questionnaires to evaluate the quality of the training and professionals' understanding of SDM. Furthermore, questionnaires will be sent by email to professionals during and after the cRCT. These will provide measures about adaptation and implementation, as well as professionals' attitudes towards the intervention and its usefulness in clinical practice.

**Table 1** Implementation domain: specific objectives, research questions and data sources and collection methods

| Implementation | Specific objectives | Research questions | Data source | Data collection |
|---|---|---|---|---|
| Fidelity and adaptation | Understand the extent to which the IMA intervention was implemented as intended. 1. How is the IMA intervention implemented? | 1.1. How consistent is the intervention implementation plan? | Operative records, website records and real-world databases (patients' clinical records). | Monitoring data extraction and questionnaires. |
| | | 1.2. Did the IMA intervention require any adaptations during the cRCT? | Professionals. | Questionnaires and field diaries. |
| Coherence | Understand how professionals make sense of the IMA intervention. 2. What is the IMA intervention for professionals? | 2.1. How is the IMA intervention conceptualised by professionals? | Professionals. | Interviews. |
| | | 2.2. What are the professionals' perspectives and attitudes towards the use and usefulness of the IMA intervention? | Professionals. | Questionnaires and interviews. |
| Cognitive participation | Understand how professionals engage and commit with the IMA intervention. 3. Who implements the IMA intervention? | 3.1. How do professionals engage and commit with the IMA intervention? | Professionals. | Questionnaires and interviews. |
| | | 3.2. What factors promote or inhibit professionals' participation and commitment? | Professionals. | Interviews. |
| Collective action | Understand how professionals make use and execute the intervention as part of their clinical practice. 4. How is the IMA intervention operationalised? | 4.1. How are the resources of the IMA intervention structured and used? | Professionals. | Questionnaires and interviews. |
| | | 4.2. To what extend and why have professionals integrated the intervention into their clinical practice? | Professionals. | Questionnaires and interviews. |
| | | 4.3. To what extent and why do participants enact the IMA intervention? | Professionals. | Questionnaires and interviews. |
| Reflexive monitoring | Understand how professionals assess and comprehend the effect of the intervention on their clinical practice. 5. How is the IMA intervention understood? | 5.1. How the professionals appraise the IMA intervention and its effects? | Professionals. | Interviews and questionnaires. |
| | | 5.2. How the professionals value the IMA intervention in comparison with standard practice? | Professionals. | Interviews and questionnaires. |

Professionals: GPs, nurses and pharmacists.
cRCT, cluster-randomised controlled trial; GPs, general practitioners; IMA, Initial Medication Adherence.

## Qualitative methods
### Field diary
A field diary will be completed by a member of the research team. It will contain field notes from periodic calls (every 2 weeks the first month, and monthly until the study finishes) to the PC centre coordinators. Additionally, field diaries will be completed by each PC centre coordinator. These will include data on any barriers, facilitators or thoughts concerning the organisation and operation of the PC centre or pharmacy and the intervention.

### Interviews
Individual semistructured interviews will be conducted during and after the cRCT with professionals to explore their perspectives and experiences after implementing the IMA intervention and with patients to determine their experience with the IMA intervention and SDM and its impact on their behaviour in relation to the treatment. Approximately 30–40 interviews will be carried out with professionals and 20–30 with patients to ensure representativeness. Focus groups will be conducted with

**Table 2** Mechanisms of impact domain: specific objectives, research questions and data sources and collection methods

| Mechanisms of impact | Specific objectives | Research questions | Data sources | Data collection |
|---|---|---|---|---|
| Participants' experiences | Understanding the mechanism of the IMA intervention that influences the outcomes and explains its logic. | 1. What are the experiences of the participants (professionals and patients) with the intervention? | Professionals and patients. | Interviews. |
| | | 2. What attitude and behaviour changes have occurred because of the intervention? | Professionals and patients. | Interviews. |
| Intended and unintended consequences | Understanding anticipated and unanticipated consequences of the IMA intervention and its effects on the outcomes. | 3. Did the intervention lead to anticipated pathways or consequences? 4. Did the intervention lead to any unanticipated pathways or consequences? | Professionals and patients. | Interviews and field diary. |

Professionals: GPs, nurses and pharmacists.
GPs, general practitioners; IMA, Initial Medication Adherence.

professionals after the cRCT to understand the intervention's impact mechanisms and explore professionals' opinions of the IMA intervention and its integration into the PC centre and pharmacy practice. Moreover, we will explore the perceived barriers and facilitators to implementation and continuity of the intervention in PC and in particular those related to COVID-19 outbreaks. About three to four focus groups will be conducted with professionals from varying PC centres. Interviews and focus groups will be recorded, anonymised and transcribed by the research team before analysis.

Different types of data will be collected at different time points: before, during, and after the trial to account for the intervention dynamics and to comprehend how the context and the intervention adapt to one another (figure 4).

## Analysis

The analysis of process evaluation data will be performed throughout the study and at the end. Quantitative data will be analysed using descriptive statistics (ie, counts, proportions and means) and regression models using Stata V.17 to describe how the intervention was implemented overall and explore variations between PC centres and pharmacies.

Qualitative data will be analysed using the principles of framework analysis by qualitative researchers.[46 47] This will help researchers to organise large amounts of data systematically and focus the analysis as a group (PC centres) and as individuals (professional and patient). Field notes (from diaries) and transcripts from the interviews will be included as narrative data. After a process of familiarisation with the data (listening to recordings and reading

**Table 3** Context domain: specific objectives, research questions and data sources and collection methods

| Context | Specific objectives | Research questions | Data sources | Data collection |
|---|---|---|---|---|
| Intervention group | Understanding the conditions in which the intervention is implemented that can be relevant to the process of the intervention mechanisms. | 1. What is the context of the PC centres? | Professionals, patients, demographic records. | Questionnaires and monitoring data extraction. |
| | | 2. What mechanisms of the IMA intervention and consequences change depending on the context, and what can explain these differences? | Professionals and patients. | Interviews and field diary. |
| | | 3. Was there any contextual factor related to the community, PC centre, professional or patient that could have influenced the outcomes of the cRCT? | Professionals. | Interviews. |
| Control group | Evaluate contextual and organisational changes, and understand the factors that could influence the process. | 4. Was there any contextual factor related to the community, PC centre, professional or patient that could have influenced the outcomes of the cRCT? | Professionals. | Interviews. |

Professionals: GPs, nurses and pharmacists.
cRCT, cluster-randomised controlled trial; GPs, general practitioners; IMA, Initial Medication Adherence; PC, primary care.

| | STUDY PERIOD | | |
|---|---|---|---|
| | Pre-trial* | Trial* | Post-trial* |
| **IMA-cRCT** | | | |
| Healthcare professionals: training | X | | |
| Healthcare professionals: implementation | | X | |
| Real-World Data (patients' clinical records): Effectiveness and Cost-Effectiveness Evaluation | | | X |
| **PROCESS EVALUATION** | | | |
| Monitoring data | ←——————————————→ | | |
| Healthcare professionals' questionnaires | X | X | X |
| Field diary | ←————→ | | |
| Healthcare professionals' interviews | | X | X |
| Patients' interviews | | X | X |

**Figure 4** Process evaluation timeline. *Pretrial: September 2021 until February 2022; trial: March 2022 until September 2022; Post-trial: September 2022 until December 2022. cRCT, cluster randomised controlled trial.

field notes), the researchers will use thematic content analysis[48] to generate a coding framework following a mixed-method approach: deductive and inductive. The coding frameworks generated by the researchers will be put in common until a final one is created and applied to all the data. Data will be organised by cases and categories and will be compared within cases (PC centres) and between cases (professionals and patients) while mapping and interpreting it. NVivo software will be used to manage the data.

### Triangulation of results

Quantitative and qualitative data from the process evaluation will be analysed separately and then interpreted in combination.[49 50] First, two researchers will combine and compare the results of both, quantitative and qualitative, analyses independently. Then, a final summary of key findings will be produced jointly by the two researchers, and if there are any unresolved disagreements, another researcher will be involved. The final summary of key findings will be presented to the rest of the research team for review and clarification. The combined interpretation of results will allow us to generate deeper insights than use of either of the methods alone.

Additionally, process and effectiveness evaluation results will be integrated. Analyses will be performed separately, and once both analyses are done, the results will be combined. Combining process and effectiveness results will facilitate better understanding and interpretation of the IMA-cRCT outcomes.

### Patient and public involvement

Patient and public were not involved in setting the research questions and outcomes of the no-initiation project, yet they have been closely involved in the development and design of the IMA intervention and its support tools and will be informed of the results through the project website suitable for a non-specialist audience.

### ETHICS AND DISSEMINATION

The IMA-cRCT and its integrated process evaluation were approved by the Research Ethics Comittee (Comitè Ètic d'Investigació amb medicaments (CEIm)) at IDIAP Jordi Gol, code CEIm 21/051 P. The IMA-cRCT is a low-intensity intervention clinical trial where groups of subjects are allocated to the intervention and control groups. Informed consent from patients participating in the clinical trial will be obtained by simplified means, and it fulfils the conditions described in Regulation (EU) No 536/2014[51] and the Real Decreto 1090/2015.[52] Details of how informed consent will be obtained by simplified means are described somewhere else.[35] Participation in the process evaluation is entirely voluntary. As approved by the CEIm, all healthcare professionals participating in the process evaluation will have signed an informed consent prior to the trial commencement agreeing to have feedback questionnaires sent by mail and to take part in an interview if invited to do so towards the end of the trial. Patients participating in the process evaluation will sign an informed consent after the recruitment and prior to the beginning of interviews. All participants have the right to refuse to participate and to withdraw from the study at any time.

Findings will be disseminated through publications and conferences, as well as presentations to healthcare professionals and stakeholders from healthcare organisations in Catalonia. Full details of the dissemination strategy are outlined in the main trial protocol.[35]

**Author affiliations**
[1]Health Technology Assessment in Primary Care and Mental Health (PRISMA) Research Group, Parc Sanitari Sant Joan de Deu, Institut de Recerca Sant Joan de Deu, St Boi de Llobregat, Catalunya, Spain
[2]Department of Paediatrics, Obstetrics, Gynaecology and Preventive Medicine, Universitat Autonoma de Barcelona, Bellaterra, Catalunya, Spain
[3]Facultat de Medicina i Ciències de la Salut, Universitat de Barcelona, Barcelona, Catalunya, Spain
[4]Research Network on Chronicity, Primary Care and Health Promotion (RICAPPS), Barcelona, Spain
[5]Unitat de Suport a la Recerca Regió Metropolitana Sud, Fundació Institut Universitari per a la recerca a l'Atenció Primària de Salut Jordi Gol i Gurina (IDIAPJGol), Barcelona, Catalunya, Spain
[6]Centro de Investigación Biomédica en Red de Epidemiología y Salud Pública (CIBERESP), Madrid, Comunidad de Madrid, Spain
[7]Primary Care Centre Bartomeu Fabrés Anglada, Institut Català de la Salut Gerència Territorial Metropolitana Sud, Barcelona, Catalunya, Spain

**Acknowledgements** This work has been carried out within the framework of the PhD programme in Methodology of Biomedical Research and Public Health at the Universitat Autònoma de Barcelona.

**Contributors** MR-V obtained financing for this study and, together with CC-P, led the design, conduct and monitoring of the process evaluation. MG-G and MTP-M supported the design of the process evaluation. All authors will have a role in collecting and analysing process evaluation data as per their field of expertise. CC-P wrote the first draft of the manuscript and AS-V, MG-G, MTP-M, CG-G, MdCO-P, IA-L and MR-V added to and approved the final manuscript.

**Funding** This project has received funding from the European Research Council under the European Union's Horizon 2020 research and innovation programme (grant agreement No 948973). IA-L has a CIBERESP contract (CIBER in Epidemiology and Public Health, CB16/02/00322). MTP-M has the '16th ICS support for the promotion of group research strategies through the intensification of researchers' (7Z20/028), from the IDIAP Jordi Gol. CG-G has the '17th ICS

support for the promotion of group research strategies through the intensification of researchers' (7Z21/019), from the IDIAP Jordi Gol.

**Competing interests** None declared.

**Patient and public involvement** Patients and/or the public were involved in the design, or conduct, or reporting, or dissemination plans of this research. Refer to the Methods section for further details.

**Patient consent for publication** Not applicable.

**Provenance and peer review** Not commissioned; externally peer reviewed.

**Open access** This is an open access article distributed in accordance with the Creative Commons Attribution 4.0 Unported (CC BY 4.0) license, which permits others to copy, redistribute, remix, transform and build upon this work for any purpose, provided the original work is properly cited, a link to the licence is given, and indication of whether changes were made. See: https://creativecommons.org/licenses/by/4.0/.

**ORCID iD**
Montserrat Gil-Girbau http://orcid.org/0000-0002-4396-6492

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
