## [Reviewer comments · BMJ Open]

ARTICLE DETAILS

TITLE (PROVISIONAL)	Complex multidisciplinary intervention to improve Initial Medication Adherence to cardiovascular disease and diabetes treatments in primary care (the IMA-cRCT study): Mixed methods process evaluation protocol
AUTHORS	Corral-Partearroyo, Carmen; Sánchez-Viñas, Alba; Gil-Girbau, Montserrat; Peñarrubia-María, Maria Teresa; Aznar-Lou, Ignacio; Gallardo-González, Carmen; Olmos-Palenzuela, María del Carmen; Rubio-Valera, Maria

VERSION 1 – REVIEW

REVIEWER	Yang, Chen Chinese University of Hong Kong Faculty of Medicine, The Nethersole School of Nursing
REVIEW RETURNED	28-Aug-2022

GENERAL COMMENTS	Thanks for inviting me to review this manuscript. This paper presents a study protocol for a mix-methods process evaluation of a complex multidisciplinary intervention. The paper is well organized and written with a rigorous study design. I have some suggestions for the authors' consideration. 1. Title: Please reconsider whether to include intervention development and implementation strategy in the title. This paper focuses on the process evaluation and only contains a few short descriptions of the intervention development and implementation strategy.2. p.3, Lines 57-58: this point seems not either the strength or limitation of the study.3. p.3, Lines 82-83: could the authors give more explanations about the risk of bias identified in previous studies? What kind of bias was identified?4. p.3, Line 84: it is not clear what a "health theory" means here. Do the authors mean behavioral change theory or cognitive behavioral theory, which may be more appropriate in the context of this study?5. p.4, Line 92: I am not sure what the "association between SDM and medication adherence" means here. Do the authors mean the effects of SDM-based interventions on medication adherence are unclear? May need to give more explanations.6. p.4, Lines 99-111: This paragraph presents the development of the study. May I know whether all of the studies mentioned in this paragraph were conducted by your research team? Also, as said before, if you would like to put intervention development in the title, I think more details should be given. A single paragraph is not enough.
---

	7. p.5, Line 133: Could the authors give more explanations about the “theoretical model for non-initiation”? Was it developed by your research team? 8. p.6: please introduce the intervention frequency and dosage. 9. p.9, Line 246: I would suggest presenting the planned sample size (study centers + patients) of the cRCT. Also, I would like to know how many centers/professionals/patients will be involved in the process evaluation. 10. pp. 9-10, Lines 261-278: I am not sure whether the description here is for the main study or process evaluation only. May indicate more clearly. 11. Tables 1-3: Most of the data sources are from professionals. May I know whether all types of health professionals (i.e., GPs, nurses, and pharmacists) will be included in addressing each research question? 12. Table 3: May I know what kind of contextual factors will be considered in this study? 13. Table 3, 4th research question: for me, patients in the control group can also be interviewed to explore the contextual factors that impact their adherence and health outcomes. The findings can be compared among patients in the intervention and control groups. What is the authors’ opinion? 14. p.16, Line 374: May I know the aim of using regression models? what research questions will be answered by the use of regression models? 15. Figure 4: I would suggest adding months (or weeks) to this timeline. It will be much clearer.
--	--

REVIEWER	Abughosh, Susan University of Houston, pharmaceutical health outcomes
REVIEW RETURNED	29-Aug-2022

GENERAL COMMENTS	The manuscript is generally well written and the methods are well described. A few suggestions are summarized below:  1. The literature describing the influence of non adherence on outcomes refers to nonadherence generally and are not specific to non-initiation which is the focus of the intervention. Is there any literature specific to non-initiation and its impact on CV outcomes? 2. There is no discussion on proposed sample size or, number of focus groups for qualitative section. 3. including a paragraph on potential limitations like potential non response bias, generalizability issues etc. would be helpful 4. a more detailed description of the specific medications targeted in the intervention is needed.
---

VERSION 1 – AUTHOR RESPONSE

Reviewer 1:

Thanks for inviting me to review this manuscript. This paper presents a study protocol for a mix-methods process evaluation of a complex multidisciplinary intervention. The paper is well organized and written with a rigorous study design. I have some suggestions for the authors’ consideration.

We thank Mr. Chen Yang for your thoughtful suggestions.

1. Title: Please reconsider whether to include intervention development and implementation strategy in the title. This paper focuses on the process evaluation and only contains a few short descriptions of the intervention development and implementation strategy.

Thank you for this suggestion. We agree with the reviewer and we have removed intervention development and implementation strategy from the title. The focus of the paper is the process evaluation protocol and we believe this title is more appropriate for this manuscript.

2. p.3, Lines 57-58: this point seems not either the strength or limitation of the study.

Thank you for noticing this. We have removed this from the strength and limitations of this study section.

3. p.3, Lines 82-83: could the authors give more explanations about the risk of bias identified in previous studies? What kind of bias was identified?

Thank you for this suggestion. We have given more explanations about the most common risk of bias identified in previous studies, p.3, lines 83-88: *“Only two of these studies reported a significant decrease in non-initiation^{15,19}, and most showed a high overall risk of bias. Hawthorne effect and desirability bias was high overall due to lack of blinding of participants and the characteristics of the outcome under study^{15-18,20}, most studies used medicine acquisition as a proxy for initiation with no further follow-up and false-positive initiation could occur when patients know they are being observed^{21,22}.”*

4. p.3, Line 84: it is not clear what a “health theory” means here. Do the authors mean behavioral change theory or cognitive behavioral theory, which may be more appropriate in the context of this study?

Thank you for this comment. We are referring to health-behaviour change theory and have clarified it in the text, p.4, lines 88-89: *“None of the interventions tested was described as being founded on a health-behaviour change theory.”*

5. p.4, Line 92: I am not sure what the “association between SDM and medication adherence” means here. Do the authors mean the effects of SDM-based interventions on medication adherence are unclear? May need to give more explanations.

Thank you for this comment. Indeed, we meant that up to now there is not sufficient evidence for a positive effect of SDM based interventions on medication adherence. Most studies showed no effect on medication adherence, yet there is a lack of standardised outcomes in studies evaluating the impact of SDM interventions on adherence to pharmacological treatments making it difficult to compare their results. We have clarified it in the text, p.4, lines 96-99: *“However, there is not sufficient evidence for an effect of SDM based interventions on medication adherence, and there is a lack of standardised outcomes in studies evaluating the impact of SDM interventions on adherence to pharmacological treatments^{23-26,29}.”*

6. p.4, Lines 99-111: This paragraph presents the development of the study. May I know whether all of the studies mentioned in this paragraph were conducted by your research team? Also, as said before, if you would like to put intervention development in the title, I think more details should be given. A single paragraph is not enough.

Thank you. The studies mentioned in this paragraph were all conducted by our research team. The team that modelled the intervention is the same that will be in charge of the process evaluation and the cRCT. The research team is familiar with the intervention and the evaluation

plan, and data collection and analysis will be conducted by quantitative and qualitative researchers. Subsequently, the insights of all the researchers will be deepened and triangulated. This process evaluation has also been presented to an external group to identify appropriate, feasible and effective methods.

Thank you for the suggestion. We agree with the editor and we have removed intervention development from the title.

7. p.5, Line 133: Could the authors give more explanations about the “theoretical model for non-initiation”? Was it developed by your research team?

Thank you for this suggestion. The theoretical model for non-initiation was developed by the research team presenting this manuscript. We agree with the reviewer's comment and have explained further the theoretical model for non-initiation so its influence on the IMA intervention is understood, p.5, lines 139-148: *“The IMA intervention is founded on the theoretical model for non-initiation^{7,33}. According to this model, the decision to initiate pharmacological treatments is multifactorial and it is influenced by the patients’ beliefs about the disease and treatment options, the existence of non-pharmacological measures, the interaction with healthcare professionals (GPs, nurses and pharmacists), and the context, cultural factors and health literacy of the patient^{7,33}. The model suggests that an intervention that improves health literacy, helping the patient to understand the risks of the disease and the benefits and risks of treatment options, and involves the patient in the decision-making process, could improve initiation and long-term adherence^{7,33}. The model also highlights the influence of healthcare professionals and the importance of multidisciplinary recommendations when a new pharmacological treatment is prescribed.”*

8. p.6: please introduce the intervention frequency and dosage.

Thank you for this suggestion. The IMA intervention is a one-shot intervention at the time of a new prescription. The dosage varies on the healthcare professionals (GPs, nurses and pharmacists) consulted during and after a new prescription and whether they are participating in the trial. This can vary between PC centres in the context of a pragmatic trial. We have clarified it in the text, p.6, lines 166-171: *“The IMA intervention is a one-shot intervention at the time of a new prescription. The dosage, or times the intervention has been applied to the same patient, varies on the healthcare professionals (GPs, nurses and pharmacists) consulted during and after a new prescription and on whether they are participating in the trial, with the minimum dose being one time (when the prescription is issued).”*

9. p.9, Line 246: I would suggest presenting the planned sample size (study centers + patients) of the cRCT. Also, I would like to know how many centers/professionals/patients will be involved in the process evaluation.

Thank you for this suggestion. We have presented in the text the planned sample size of the cRCT as well as referenced the cRCT protocol for further detail, p.10, line 272: *“The IMA-cRCT will recruit 24 PC centres, around 300 professionals and 4000 patients³⁵.”*

In addition, we have included information on the proposed number of centres, professionals and patients that will be involved in the process evaluation. Professionals from all PC centres in the intervention group will have feedback questionnaires sent to them, but only a selection of PC centres, professionals and patients will be invited to participate in the interviews due to time restrictions. Numbers have been estimated to ensure representativeness of PC centres based on the type of PC centres (size and location, socioeconomic level of the area, rurality, and the proportion of immigrant population).

Setting and participants, Intervention group, Professionals, p.10, line 277-282: “GPs, nurses and pharmacists from all PC centres will have feedback questionnaires sent to them, but only professionals from 6 to 8 PC centres will be recruited to participate in the qualitative research due to time restrictions. The recruitment will be based on a theoretical sampling strategy according to the type of PC centre (size and location), and taking into account predictors of non-initiation (socioeconomic level of the area, rurality, and the proportion of immigrant population).

Data Collection Methods, Qualitative Methods, Interviews, p.17, lines 383-385: “Approximately 30 to 40 interviews will be carried out with professionals and 20 to 30 with patients to ensure representativeness.”, and p.17, lines 389-390: “About 3 to 4 focus groups will be conducted with professionals from varying PC centres.”

10. pp. 9-10, Lines 261-278: I am not sure whether the description here is for the main study or process evaluation only. May indicate more clearly.

Thank you for this comment. The description refers to the participants of the process evaluation. We have clarified this point in the text, p.10, lines 272-274: “The sampling strategy of the process evaluation is conditioned by the cluster sampling design of the cRCT and is detailed below.”

11. Tables 1-3: Most of the data sources are from professionals. May I know whether all types of health professionals (i.e., GPs, nurses, and pharmacists) will be included in addressing each research question?

Thank you for this comment. Indeed, all types of healthcare professionals; GPs, nurses and pharmacists will be included in addressing the research questions. Interviews with GPs will explore their role as prescribers in the IMA intervention, and interviews with nurses and pharmacists will explore their role as key supporters in the IMA intervention. We have clarified it in the table legends, p.13 line 323, p.14 line 333, p.15 line 342: “Professionals: GPs, nurses and pharmacists.”

12. Table 3: May I know what kind of contextual factors will be considered in this study?

Thank you for this comment. We will explore the context to understand the conditions in which the intervention is actually implemented and those factors that can influence the study outcomes in the framework of a pragmatic trial in which the context may vary between PC centres and pharmacies as detailed in the text (p.14, lines 334-339). For instance, we will explore how the COVID-19 pandemic has affected clinical practice in PC centres and pharmacies, patient access to primary care centres, professional workload, professionals transferred to other centres and other interventions implemented during the time of the trial that could have influenced adherence to medications.

13. Table 3, 4th research question: for me, patients in the control group can also be interviewed to explore the contextual factors that impact their adherence and health outcomes. The findings can be compared among patients in the intervention and control groups. What is the authors' opinion?

Thank you for this. Contextual factors that impact initiation and adherence for patients receiving usual care were explored during phase 1 of the non-initiation project. Patients from primary care centres in Catalonia were interviewed to explore patients' reasons for non-initiation as well as contextual factors that influence their behaviour towards initiation and adherence (Gil-Girbau et al., 2020). Contextual factors of PC centres during the cRCT explained above will be explored with professionals as we believe they will have richer information about it.

Gil-Girbau M, Aznar-Lou I, Peñarrubia-María MT, Moreno-Peral P, Fernández A, Bellón JÁ, et

al. Reasons for medication non-initiation: A qualitative exploration of the patients' perspective. *Res Social Adm Pharm.* 2020 May;16(5):663–72.

14. p.16, Line 374: May I know the aim of using regression models? what research questions will be answered by the use of regression models?

Thank you for this comment. Regression models will be used to assess variations between primary care centres on professional engagement, training attendance and adaptation and implementation of the IMA intervention, taking into account demographic characteristics of PC centres as independent variables (size and location, number of professionals, socioeconomic level of the area, rurality, average age of the population and proportion of immigrant population). This is explained on p.17 line 399.

15. Figure 4: I would suggest adding months (or weeks) to this timeline. It will be much clearer.

Thank you for this suggestion. The figure legend now includes the months of the timeline, p.26, lines 654-655: “*Pre-trial: September 2021 until February 2022; Trial: March 2022 until September 2022; Post-trial: September 2022 until December 2022.”

Reviewer 2:

The manuscript is generally well written and the methods are well described. A few suggestions are summarized below.

We thank Dr. Susan Abughosh for your thoughtful suggestions.

1. The literature describing the influence of non adherence on outcomes refers to nonadherence generally and are not specific to non-initiation which is the focus of the intervention. Is there any literature specific to non-initiation and its impact on CV outcomes?

Thank you for this comment. To date, there is little existing literature on non-initiation and its impact on CV outcomes. One of the papers identified refers specifically to non-initiation after a cardiovascular event (Jackevicius, Li and Tu, 2008) and two after a coronary procedure (Ko et al., 2009; Cruden et al., 2014). Lastly, two studies evaluated clinical parameters after non-initiation of antidiabetic and antihypertensive medications, both studies showed a greater improvement in patients who initiate, but initiators had worse clinical parameters at the time of prescription (Shah et al., 2009; Shah et al., 2009).

The IMA intervention's primary focus is non-initiation, yet secondary adherence is expected to improve and it is a secondary outcome of the intervention. Therefore, the authors believe that the literature presented in the paper about the impact of non-adherence generally on health outcomes may be used as a justification for designing and implementing interventions to improve patients' health outcomes in the long run.

Jackevicius, C.A., Li, P. and Tu, J. V. (2008) 'Prevalence, predictors, and outcomes of primary nonadherence after acute myocardial infarction', *Circulation*, 117(8), pp. 1028–1036.

Ko, D.T. *et al.* (2009) 'Patterns of use of thienopyridine therapy after percutaneous coronary interventions with drug-eluting stents and bare-metal stents', *American Heart Journal*, 158(4), pp. 592-598.e1.

Cruden, N.L. *et al.* (2014) 'Delay in filling first clopidogrel prescription after coronary stenting is associated with an increased risk of death and myocardial infarction', *Journal of the American Heart Association*, 3(3), pp. 1–7.

Shah, N.R., Hirsch, A.G., Zacker, C., Taylor, S., *et al.* (2009) 'Factors associated with first-fill adherence rates for diabetic medications: A cohort study', *Journal of General Internal Medicine*, 24(2), pp. 233–237.

Shah, N.R., Hirsch, A.G., Zacker, C., Wood, G.C., *et al.* (2009) 'Predictors of First-Fill Adherence for Patients With Hypertension', *American journal of hypertension*, 22(4), p. 392.

2. There is no discussion on proposed sample size or, number of focus groups for qualitative section.

Thank you for this suggestion. We have presented in the text the planned sample size of the cRCT as well as referenced the cRCT protocol for further detail, p.10, line 272: *"The IMA-cRCT will recruit 24 PC centres, around 300 professionals and 4000 patients³⁵."*

In addition, we have included information on the proposed number of centres, professionals and patients that will be involved in the process evaluation. Professionals from all PC centres in the intervention group will have feedback questionnaires sent to them, but only a selection of PC centres, professionals and patients will be invited to participate in the interviews due to time restrictions. Numbers have been estimated to ensure representativeness of all type of PC centres (size and location, socioeconomic level of the area, rurality, and the proportion of immigrant population).

Setting and participants, Intervention group, Professionals, p.10, line 277-282: *"GPs, nurses and pharmacists from all PC centres will have feedback questionnaires sent to them, but only professionals from 6 to 8 PC centres will be recruited to participate in the qualitative research due to time restrictions. The recruitment will be based on a theoretical sampling strategy according to the type of PC centre (size and location), and taking into account predictors of non-initiation (socioeconomic level of the area, rurality, and the proportion of immigrant population)."*

Data Collection Methods, Qualitative Methods, Interviews, p.17, lines 383-385: *"Approximately 30 to 40 interviews will be carried out with professionals and 20 to 30 with patients to ensure representativeness."*, and p.17, lines 389-390: *"About 3 to 4 focus groups will be conducted with professionals from varying PC centres."*

3. including a paragraph on potential limitations like potential non-response bias, generalizability issues etc. would be helpful

Thank you for this suggestion. A paragraph on strengths and limitations was not included in the text as per the journal guidelines. However, we agree with the reviewer's comment and have added a new bullet point on the strengths and limitations of this study section with the main potential biases identified. In this section, we have tried to focus on the methods and not the results of the study as indicated by the journal guidelines, p.3, lines 55-70.

"Strengths and limitations of this study"

- *This process evaluation will explain how the intervention was implemented, how different components interact and work, and how they influence outcomes.*
- *This study includes a wide range of quantitative and qualitative research methods, it is logistically challenging and time-consuming. A multidisciplinary research team has been involved.*
- *The flexible and pragmatic design will be crucial to react to changes and adapt the intervention to emerging contextual factors.*
- *Data collection methods have been designed to adapt to the participants in what we anticipate might be an overloaded and difficult time due to the persisting COVID-19 pandemic.*

- *There is a risk of response bias among professionals that answer questionnaires and agree to participate in the qualitative evaluation as they may have engaged more with the intervention. Additionally, patients will be recruited by professionals and this might bias their responses and the decision of the patient towards filling the prescription.”*

4. a more detailed description of the specific medications targeted in the intervention is needed.

Thank you for this suggestion. A more detailed description of the specific medications targeted by the intervention has been added to the text and the study protocol has been referenced for further details, p.5, lines 132-135: *“Professionals in the intervention group were trained on the IMA intervention and will apply it to all patients receiving a new prescription for lipid-lowering medication, antihypertensive medication, anti-platelet medication and/or antidiabetic medication during the study period (7 months)³⁵.”*

VERSION 2 – REVIEW

REVIEWER	Yang, Chen Chinese University of Hong Kong Faculty of Medicine, The Nethersole School of Nursing
REVIEW RETURNED	07-Oct-2022
GENERAL COMMENTS	The authors have addressed all my comments. I have no further comments. Thank you.